# The relationship between NLR, LDL-C/HDL-C, NHR and coronary artery disease

**Shuaishuai Yuan[1]◉, Lingling Li[2]◉¤, Tian Pu[3]◉, Xizhen Fan[4], Zheng Wang[3], Pailing Xie[1]\*, Peijun Li[1]\***

1 Division of Cardiovascular Intensive Care (C-ICU), Cardiac and Vascular Center, The University of Hong Kong-Shenzhen Hospital, Shenzhen, Guangdong, China, 2 China Medical University, Shenyang, Liaoning, China, 3 The First Affiliated Hospital of Anhui Medical University, Hefei, Anhui, China, 4 Division of Life Science and Medicine, The First Affiliated Hospital of USTC, University of Science and Technology of China, Hefei, Anhui, China

◉ These authors contributed equally to this work.
¤ Current address: Shenzhen Maternity and Child Healthcare Hospital, Shenzhen, Guangdong, China
\* pauling121@163.com (PX); 13213573655@163.com (PL)

**Data Availability Statement:** All relevant data are within the manuscript and its Supporting Information files.

**Funding:** The study was supported by the general program of the National Natural Science

## Abstract

### Objective

Chronic inflammation and dyslipidemia are key risk factors for atherosclerotic cardiovascular diseases. We retrospectively explored the association between the neutrophil to lymphocyte ratio (NLR), the ratio of low-density lipoprotein cholesterol (LDL-C) to high-density lipoprotein cholesterol (HDL-C), and the neutrophil to HDL-C ratio (NHR), and the severity of coronary lesions in patients with acute coronary syndrome (ACS).

### Method

In June 2023, we selected 1210 patients who were diagnosed with ACS based on chest pain from January 2017 to December 2022. Of these, 1100 patients with abnormal coronary angiography were categorized into the experimental group, and 110 patients with normal coronary angiography were classified as the control group. We collected routine blood tests, lipid profiles, and coronary angiography results at admission (before coronary angiography). Patients were then stratified into a control group (Gensini score = 0) and an experimental group (Gensini score = 0) based on the Gensini score. The experimental group was further divided into a low score group (Gensini score < 69) and a high score group (Gensini score $\geq$ 69).

### Result

1. Statistically significant differences were observed between the control and experimental groups in terms of gender, age, body mass index (BMI), hypertension, diabetes, smoking history, and counts of neutrophils (NEU), lymphocytes (LYM), monocytes (MON), eosinophils (EOS), red cell distribution width (RDW), total cholesterol (TC), HDL-C, LDL-C, NLR, LDL-C/HDL-C, and NHR (P<0.05). Furthermore, differences in BMI, hypertension, diabetes, smoking history, NEU, LYM, MON, TC, triglyceride (TG), HDL-C, LDL-C, NLR, LDL-C/HDL-C, and NHR were significant between the low and high score groups (P<0.05). 2.

Foundation of China (82170435). The funders had no role in study design, data collection and analysis, decision to publish, or preparation of the manuscript.

**Competing interests:** The authors have declared that no competing interests exist.

**Abbreviations:** NLR, Neutrophil to lymphocyte ratio; LDL-C, Low-density lipoprotein cholesterol; HDL-C, High-density lipoprotein cholesterol; NHR, Neutrophil to high-density lipoprotein cholesterol ratio; ACS, Acute coronary syndrome; BMI, Body mass index; NEU, Neutrophil count; LYM, Lymphocyte count; MON, Monocyte count; EOS, Eosinophil count; RDW, Red cell distribution width; ROC, Receiver operating characteristic curve; TC, Total cholesterol; TG, Triglyceride; CAD, Coronary artery disease; BAS, Basophil count; RBC, Red blood cell count; MCV, Mean corpuscular volume; SRs, Macrophages express scavenger receptors; SR-A1, A1 Scavenger receptor; CD36, Cluster of differentiation 36; LOX-1, Hemagglutinin oxidized low-density lipoprotein receptor-1.

NEU, LYM, MON, TC, HDL-C, LDL-C, NLR, LDL-C/HDL-C, and NHR showed significant correlations with the Gensini score (r>0.2, P<0.05), with NLR and LDL-C/HDL-C showing the strongest correlations (r = 0.822, P = 0.000). 3. The Receiver Operating Characteristic (ROC) curve indicated that the combination of NLR and LDL-C/HDL-C had superior sensitivity and specificity in predicting the severity of coronary lesions, with a significant difference (P<0.05). The sensitivity was 87.1%, the specificity was 90.9%, and the cut-off point was 2.04. 4. A predictive model was developed based on the ratio of NLR and LDL-C/HDL-C to the Gensini score. The final model score was calculated as 6.803 + 7.029NLR + 13.079LDL-C/HDL-C ($R^2$ = 0.708).

## Conclusion

Compared to NLR, LDL-C/HDL-C, and NHR, the combined NLR and LDL-C/HDL-C ratio is a more accurate marker for assessing the severity of coronary artery disease in ACS patients. Its convenience and effectiveness make it a promising tool for early assessment, timely risk stratification, and appropriate clinical intervention, ultimately improving clinical outcomes for ACS patients.

## Introduction

Coronary artery disease (CAD) is a devastating condition characterized by high incidence and mortality rates [1]. Each year in Europe, over a million patients are hospitalized due to acute coronary syndrome (ACS), leading to millions of deaths globally [2]. CAD is increasingly recognized as a primary cause of death worldwide [3]. Atherosclerosis, driven by inflammation and abnormalities in lipid metabolism, plays a central role in the disease's pathophysiology [4].

The release of inflammatory cytokines and the overall inflammatory response are key pathophysiological factors in coronary atherosclerosis [5]. Research has shown that elevated levels of inflammatory biomarkers correlate with an increased risk of cardiac events in CAD patients [6]. Recently, NLR has been highlighted as a significant marker in the progression of ACS and as a predictor of the severity of coronary artery lesions [7]. Dong CH et al. demonstrated that NLR could predict mortality and major adverse cardiac events in ACS [8]. Additionally, dyslipidemia, marked by increased levels of TG, TC, LDL-C, and HDL-C, escalates the risk of atherosclerosis [9]. Compared to LDL-C or HDL-C alone, the LDL-C/HDL-C ratio offers a more precise measure of coronary lesion severity and cardiovascular risk [10, 11]. In recent years, NHR has also garnered interest as a promising predictor of coronary lesion severity and a tool for risk stratification in cardiovascular medicine [4, 12].

Both inflammation and abnormalities in lipid metabolism are crucial in the development of atherosclerosis. However, studies combining these two factors are scarce, and the importance of their integration has not been fully recognized. This study proposes that combining inflammatory markers with lipid profiles provides a more accurate assessment of coronary lesion severity in ACS patients. This approach is straightforward, cost-effective, widely accessible, and non-invasive, making it an invaluable tool for evaluating ACS patients and guiding timely clinical interventions, especially in settings with limited healthcare resources.

## Data and methods

### Research subjects

In June 2023, we selected 1,210 patients diagnosed with ACS for chest pain from three institutions: The University of Hong Kong Shenzhen Hospital, The First Affiliated Hospital of the University of Science and Technology of China, and The First Affiliated Hospital of Anhui Medical University. The timeframe for patient selection was January 2017 to December 2022. Diagnosis adhered to the criteria from the Fourth Universal Definition of Myocardial Infarction published in 2018 [13]. The inclusion criteria were: (1) A clinical presentation of myocardial ischemia lasting over 20 minutes, unrelieved by nitroglycerin; (2) New-onset ST-segment elevation of $\geq 0.1$ mV or pathological Q-wave on the electrocardiogram; (3) Elevated serum cardiac biomarker levels, specifically troponin. The exclusion criteria included: (1) Recent infectious diseases; (2) Use of antimicrobial drugs in the past two weeks; (3) Acute cardiovascular diseases in the past six months; (4) History of myocardial infarction, malignant tumors, immune disorders, hematological disorders, or severe hepatic or renal dysfunction; (5) Recent use of immunosuppressive drugs; (6) Recent history of blood transfusion or acute stress; (7) Prior use of lipid-lowering drugs. A total of 1,100 patients with abnormal coronary angiography findings were grouped into the experimental group, and 110 with normal findings were placed in the control group. This retrospective study received approval from the Ethics Committee, was granted exemption from informed consent, and adhered to the Declaration of Helsinki (ethical codes: 2020-P-062, [2023] 109). There were no competing interests, and all procedures followed the relevant guidelines.

### Data collection

We documented the following patient data: gender, age, body mass index (BMI), and medical history (smoking, alcohol consumption, hypertension, diabetes). Upon admission, we collected routine blood indicators in the emergency department before coronary angiography: NEU, LYM, NLR, MON, BAS, EOS, RBC, MCV, and RDW. Lipid profiles were also obtained, including TC, TG, LDL-C, and HDL-C. The results of coronary angiography were recorded. The severity of CAD was assessed using the Gensini score [14]. Patients were divided into two groups: a control group (n = 110) with a Gensini score of 0 and an experimental group (n = 1,100) with a Gensini score greater than 0. Patients in the experimental group were further categorized into a low-score group (Gensini score < 69) and a high-score group (Gensini score $\geq$ 69). The Gensini score was independently calculated by two physicians; discrepancies were resolved by a third physician (chief physician).

### Statistical methods

The data were analyzed using SPSS 22.0 statistical software. The Kolmogorov-Smirnov test was used to assess the normality of continuous variables. Normally distributed variables are presented as mean ± standard deviation, whereas non-normally distributed variables are expressed as medians. For comparing quantitative data between two groups, the independent sample t-test and the rank-sum test were utilized; categorical data were analyzed using the $\chi^2$ test. The relationship between each variable and the Gensini score was determined through Pearson correlation analysis, denoted by the correlation coefficient r. Binary logistic regression analysis was applied to NLR and LDL-C/HDL-C to identify combined diagnostic factors. The predictive power was assessed using the receiver operating characteristic (ROC) curve. A predictive model was developed using a stepwise approach in multiple-factor linear regression analysis, with variables entered at a significance level of 0.05 and removed at 0.10. Statistical significance was established at $p < 0.05$.

## Results

Baseline characteristics (Tables 1 and 2): The study included 1,100 ACS patients, with 824 males. Significant differences were observed between the control and experimental groups in terms of gender, age, BMI, hypertension, diabetes, smoking history, NEU, LYM, MON, EOS, RDW, TC, HDL-C, LDL-C, NLR, LDL-C/HDL-C, and NHR (p<0.05). Additionally, significant differences were noted between the low-score and high-score groups in terms of BMI, hypertension, diabetes, smoking history, NEU, LYM, MON, TC, TG, HDL-C, LDL-C, NLR, LDL-C/HDL-C, and NHR (p<0.05).

Pearson correlation analysis (Table 3) was utilized to evaluate the associations between various clinical parameters—age, BMI, NEU, LYM, MON, EOS, RDW, NLR, TC, HDL-C, LDL-C, LDL-C/HDL-C, NHR—and the Gensini score in the experimental group. The analysis revealed significant correlations for BMI, NEU, LYM, MON, TC, HDL-C, LDL-C, NLR, LDL-C/HDL-C, and NHR with the Gensini score (r > 0.2, p < 0.05). Notably, NLR and LDL-C/HDL-C showed the strongest associations (r > 0.4). Binary logistic regression analysis on NLR and LDL-C/HDL-C was conducted to derive a combined diagnostic factor, represented as NLR&LDL-C/HDL-C = -8.842 + 2.449NLR + 2.877LDL-C/HDL-C. This factor demonstrated a robust correlation with the Gensini score (r = 0.822, P = 0.000), outperforming the individual ratios of NLR and LDL-C/HDL-C.

The predictive value of NLR, LDL-C/HDL-C, NHR, and NLR&LDL-C/HDL-C for the severity of coronary artery disease was assessed (Table 4; Fig 1). The area under the receiver operating characteristic curve (ROC) for NLR was 0.830, with a sensitivity of 78.3%, specificity of 84.5%, and a cutoff value of 2.00. For LDL-C/HDL-C, the area under the curve (AUC) was 0.805, achieving a sensitivity of 77.4%, specificity of 82.7%, and a cutoff of 2.14. The AUC for

**Table 1. Comparison of baseline data between the control group and case group.**

| Groups | Control group(Gensini = 0,n = 110) | Case group(Gensini>0,n = 1100) | t/x²/z | P |
|---|---|---|---|---|
| Male,n(%) | 68(61.8%) | 824(74.9%) | 8.8 | 0.003 |
| Hypertension,n(%) | 44(40.0%) | 618(56.2%) | 10.6 | 0.001 |
| Diabetes mellitus,n(%) | 36(32.7%) | 542(49.3%) | 10.9 | 0.001 |
| Smoking,n(%) | 41(37.3%) | 556(50.5%) | 7.0 | 0.008 |
| Drinking, n(%) | 59(53.6%) | 687(62.5%) | 3.3 | 0.070 |
| Age, years | 55.0±11.7 | 61.7±13.2 | -5.10 | 0.000 |
| BMI(kg/m^2) | 21.8±2.7 | 23.5±2.5 | -6.5 | 0.000 |
| NEU(10^9/L) | 3.6±1.0 | 5.6±1.9 | -18.4 | 0.000 |
| LYM(10^9/L) | 2.3±0.6 | 1.8±0.6 | 5.7 | 0.000 |
| MON(10^9/L) | 0.4±0.2 | 0.6±0.3 | -10.4 | 0.000 |
| BAS(10^9/L) | 0.0±0.1 | 0.1±0.1 | -0.7 | 0.457 |
| EOS(10^9/L) | 0.1±0.1 | 0.1±0.1 | 2.4 | 0.018 |
| RBC(10^12/L) | 4.5±0.6 | 4.4±0.7 | 1.6 | 0.121 |
| MCV(fL) | 90.8±4.7 | 90.7±5.4 | 0.3 | 0.754 |
| RDW(%) | 12.6±1.0 | 13.2±1.2 | -3.2 | 0.002 |
| TC(mmol/l) | 3.7±1.0 | 4.4±1.1 | -6.5 | 0.000 |
| TG(mmol/l) | 1.5±1.0 | 1.7±1.2 | -1.3 | 0.209 |
| HDL-C(mmol/l) | 1.1±0.4 | 1.0±0.3 | 4.4 | 0.000 |
| LDL-C(mmol/l) | 1.8±0.7 | 2.7±0.9 | -11.0 | 0.000 |
| NLR | 1.7±0.6 | 3.3±2.1 | -19.2 | 0.000 |
| LDL-C/HDL-C | 1.7±0.6 | 2.9±1.2 | -18.4 | 0.000 |
| NHR | 3.4±1.3 | 6.2±2.7 | -18.5 | 0.000 |

**Table 2. Comparison of baseline data between low rating group and high rating group.**

| Groups | Low rating group(Gensini<69,n = 514) | High rating group(Gensini≥69,n = 586) | t/x²/z | P |
|---|---|---|---|---|
| Male,n(%) | 380(73.9%) | 444(75.8%) | 0.5 | 0.483 |
| Hypertension,n(%) | 264(51.4%) | 354(60.4%) | 9.1 | 0.003 |
| Diabetes mellitus,n(%) | 235(45.7%) | 307(52.4%) | 4.9 | 0.027 |
| Smoking,n(%) | 237(46.1%) | 319(54.4%) | 7.6 | 0.006 |
| Drinking, n(%) | 310(60.3%) | 377(64.3%) | 1.9 | 0.169 |
| Age, years | 61.5±12.8 | 61.8±13.5 | -0.4 | 0.698 |
| BMI(kg/m^2) | 23.0±2.5 | 23.5±2.5 | -6.0 | 0.000 |
| NEU(10^9/L) | 4.7±1.3 | 6.4±2.0 | -16.8 | 0.000 |
| LYM(10^9/L) | 2.1±0.6 | 1.7±0.6 | 11.7 | 0.000 |
| MON(10^9/L) | 0.5±0.2 | 0.6±0.3 | -8.5 | 0.000 |
| BAS(10^9/L) | 0.1±0.1 | 0.0±0.1 | 0.7 | 0.481 |
| EOS(10^9/L) | 0.1±0.1 | 0.1±0.1 | 0.9 | 0.388 |
| RBC(10^12/L) | 4.4±0.7 | 4.4±0.7 | 0.3 | 0.793 |
| MCV(fL) | 90.8±5.4 | 90.5±5.5 | 0.9 | 0.368 |
| RDW(%) | 13.2±1.0 | 13.3±1.4 | -0.6 | 0.554 |
| TC(mmol/l) | 4.0±0.9 | 4.8±1.0 | -14.8 | 0.000 |
| TG(mmol/l) | 1.6±1.1 | 1.8±1.3 | -2.1 | 0.040 |
| HDL-C(mmol/l) | 1.0±0.3 | 0.9±0.2 | 7.1 | 0.000 |
| LDL-C(mmol/l) | 2.2±0.7 | 3.0±0.9 | -16.5 | 0.000 |
| NLR | 2.4±1.0 | 4.2±2.4 | -16.6 | 0.000 |
| LDL-C/HDL-C | 2.2±0.7 | 3.5±1.2 | -21.1 | 0.000 |
| NHR | 4.9±1.9 | 7.4±2.8 | -17.4 | 0.000 |

NHR stood at 0.800, with a sensitivity of 69.5%, specificity of 86.4%, and a cutoff of 4.61. The AUC for NLR&LDL-C/HDL-C reached 0.922, with a sensitivity of 87.1%, specificity of 90.9%, and a cutoff of 2.04. Among these predictors, NLR&LDL-C/HDL-C displayed the highest sensitivity and specificity in determining the severity of coronary artery disease, with a statistically significant difference (p<0.05).

A multivariable linear regression analysis was performed on NLR and LDL-C/HDL-C in ACS patients (Table 5). Using stepwise regression, the relationship between the NLR, LDL-C/

**Table 3. Correlation analysis of blood routine spectrum, blood lipid spectrum, and Gensini score.**

| Item | r | P |
|---|---|---|
| Age, years | 0.016 | 0.601 |
| BMI(kg/m^2) | 0.173 | 0.000 |
| NEU(10^9/L) | 0.530 | 0.000 |
| LYM(10^9/L) | -0.403 | 0.000 |
| MON(10^9/L) | 0.248 | 0.000 |
| EOS(10^9/L) | -0.034 | 0.256 |
| RDW(%) | 0.043 | 0.154 |
| TC(mmol/l) | 0.417 | 0.000 |
| HDL-C(mmol/l) | -0.265 | 0.000 |
| LDL-C(mmol/l) | 0.502 | 0.000 |
| NLR | 0.593 | 0.000 |
| LDL-C/HDL-C | 0.623 | 0.000 |
| NHR | 0.606 | 0.000 |

**Table 4. ROC analysis area diagram of NLR, LDL-C/HDL-C, NHR, NLR&LDL-C/HDL-C.**

| | Area | Std. Error[a] | Asymptotic Sig.[b] | Asymptotic 95% Confidence Interval | |
| --- | --- | --- | --- | --- | --- |
| | | | | Lower Bound | Upper Bound |
| NEU | 0.849 | 0.016 | 0.000 | 0.816 | 0.881 |
| LYM | 0.667 | 0.027 | 0.000 | 0.614 | 0.721 |
| NLR | 0.870 | 0.017 | 0.000 | 0.836 | 0.903 |
| LDL-C | 0.785 | 0.021 | 0.000 | 0.745 | 0.826 |
| HDL-C | 0.665 | 0.025 | 0.000 | 0.615 | 0.715 |
| LDL-C/HDL-C | 0.857 | 0.016 | 0.000 | 0.825 | 0.888 |
| NHR | 0.838 | 0.017 | 0.000 | 0.805 | 0.870 |
| NLR&LDL-C/HDL-C | 0.950 | 0.009 | 0.000 | 0.933 | 0.966 |

HDL-C ratio, and the Gensini score was explored. The resultant predictive model for the Gensini score was expressed as follows: Gensini score = 6.803 + 7.029NLR + 13.079LDL-C/HDL-C ($R^2$ = 0.708).

## Discussion

The results of this study confirmed that elevated levels of NLR and LDL-C/HDL-C are strongly linked to the severity of coronary artery stenosis. Our research introduces NLR and LDL-C/HDL-C as a new, simple, and accessible index, combining inflammatory markers with lipid profiles, which may aid in predicting CAD and assessing coronary artery stenosis.

In cardiovascular medicine, the high incidence and mortality of ACS underscore the need to evaluate the severity of coronary artery stenosis in patients presenting with chest pain. This evaluation is essential for determining appropriate treatment strategies and cardiovascular risk

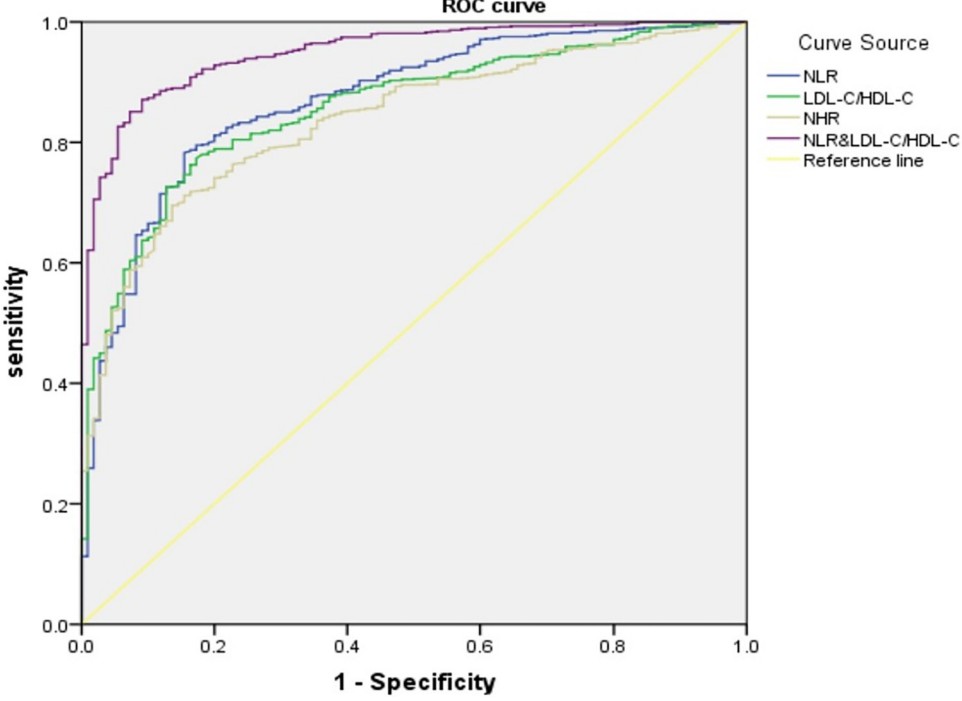

**Fig 1. The ROC analysis of NLR, LDL-C/HDL-C, NHR, NLR&LDL-C/HDL-C.**

**Table 5. Multivariate linear regression analysis of NLR, LDL-C/HDL-C, and Gensini score.**

| Model | | Unstandardized Coefficients | | Standardized Coefficients | t | Sig. | 95.0% Confidence Interval for B | |
|---|---|---|---|---|---|---|---|---|
| | | B | Std. Error | Beta | | | Lower Bound | Upper Bound |
| 1 | (Constant) | 28.775 | 1.605 | | 17.926 | 0.000 | 25.625 | 31.925 |
| | LDL-C/HDL-C | 13.616 | 0.516 | 0.623 | 26.364 | 0.000 | 12.602 | 14.629 |
| 2 | (Constant) | 6.803 | 1.275 | | 5.334 | 0.000 | 4.301 | 9.305 |
| | LDL-C/HDL-C | 13.079 | 0.357 | 0.598 | 36.664 | 0.000 | 12.379 | 13.779 |
| | NLR | 7.029 | 0.202 | 0.567 | 34.767 | 0.000 | 6.633 | 7.426 |

stratification [15]. There is growing evidence that inflammation and lipid abnormalities are significant risk factors for ACS and coronary atherosclerosis [16–18]. The findings suggest that critical values for NLR are above 2.00, for LDL-C/HDL-C above 2.14, and for NHR above 4.61. A critical value for the combined index NLR&LDL-C/HDL-C above 2.04 has shown good sensitivity and specificity in predicting the presence of coronary artery atherosclerosis prior to coronary angiography. The NLR&LDL-C/HDL-C index, a novel composite biomarker of inflammatory cells and blood lipids, may provide a more comprehensive assessment than single biomarkers. Both univariate and multivariate analyses demonstrate that NLR&LDL-C/ HDL-C has a stronger correlation with the Gensini score than NHR, suggesting it better reflects the severity of coronary artery lesions in patients.

Inflammation is a key driver in the entire atherosclerotic process, with neutrophils playing a crucial role. Neutrophils infiltrate the vasculature in the early stages of vascular sclerosis [19]. They enhance the inflammatory response by increasing macrophage and antigen levels. Additionally, neutrophils contribute to acute tissue damage by secreting inflammatory mediators [20], primarily through the release of pro-inflammatory cytokines and reactive oxygen species, which initiate the inflammatory response. This mechanism may also involve a reduction in nitric oxide utilization, leading to endothelial dysfunction and subsequent atherosclerosis formation [21]. Furthermore, the body releases hormones like catecholamines that regulate inflammatory substance levels in the blood, resulting in a decreased lymphocyte count. In ACS patients, stress and other factors can reduce lymphocyte levels in the plasma, significantly decreasing the CD4/CD8 ratio, weakening immune function, and inducing a short-term inflammatory state [22]. This contributes to arterial sclerosis development.

During the inflammatory process, macrophages and regulatory T lymphocytes secrete anti-inflammatory molecules such as transforming growth factor-beta (TGF-beta) and interleukin-10 (IL-10), which can inhibit the inflammatory response and prevent thrombus formation. Moreover, molecules like TGF-beta, in conjunction with IL-17A, can promote plaque fibrosis and stabilization, as well as participate in tissue repair processes [23, 24]. NLR provides a comprehensive reference for the immune pathway and is more predictive of inflammation in ACS than individual neutrophils or lymphocytes [25, 26]. Studies by Guclu K [27] and Uysal HB [28] have shown the predictive value of inflammatory parameters in non-ST-elevation acute coronary syndrome patients and the severity of coronary artery disease, respectively. Our study demonstrated that patients in the ACS experimental group exhibited significantly elevated NLR compared to the control group, aligning with findings by Shumilah AM and Li X [29, 30].

Abnormal blood lipid levels also pose risks for coronary atherosclerosis and cardiovascular outcomes [31]. Increased serum LDL-C levels and decreased HDL-C levels are major risk factors for coronary atherosclerosis [32, 33]. For every 1.0 mmol/L decrease in LDL-C, cardiovascular mortality and non-fatal myocardial infarction decrease by 20–25%. Current guidelines in Europe and the United States recommend an LDL-C treatment target of less than 70 mg/dL

(1.8 mmol/L) [34]. The homeostasis of LDL-C is maintained by macrophages, which express scavenger receptors (SRs) such as SR-A1, CD36, and LOX-1. These receptors participate in the reverse transport of cholesterol to lysosomes for processing into free cholesterol, which is then excreted from the cell via cholesterol transporters. In atherosclerosis, pro-inflammatory stimuli increase the expression of LOX-1 and decrease the expression of cholesterol transport proteins, leading to the accumulation of cholesterol in macrophages and the formation of foam cells. Concurrently, LDL-C can enter blood vessels, prompting the release of adhesion molecules and chemotactic factors from endothelial cells, which contribute to foam cell formation and the development of early plaques (fatty streaks) [35].

HDL-C inhibits these processes and has been shown to suppress the expression of vascular cell adhesion molecule-1 in endothelial cells, playing an anti-atherosclerotic role [36]. Additionally, HDL-C, due to its protein and phospholipid composition, has anti-inflammatory effects and delays the progression of atherosclerosis [37]. The LDL-C/HDL-C ratio is a more accurate predictor of lipid levels in ACS than either LDL-C or HDL-C alone [38]. Ting Sun et al. [39] demonstrated the predictive value of the LDL-C/HDL-C ratio in coronary atherosclerotic heart disease. Zhixiong Zhong et al. [40] found a correlation between a high LDL-C/HDL-C ratio and cardiovascular events in ACS patients. In this study, we observed that patients in the ACS experimental group had significantly higher LDL-C/HDL-C levels compared to the control group, consistent with findings by Po Gao et al. [41].

The subject of the neutrophil to high-density lipoprotein cholesterol ratio (NHR) has also garnered attention. Kou T, Gao J, et al. [42, 43], among others, have described NHR as not only closely related to coronary artery stenosis but also as an independent predictor of severe coronary artery stenosis. Lamichhane P et al. [44] have demonstrated that NHR can be used for risk stratification and for predicting short-term and long-term prognoses in clinical cardiovascular diseases. Additionally, Liu M et al. [45] identified an independent correlation between NHR and coronary heart disease and severe stenosis in patients presenting with chest pain. NHR also showed comparable predictive value for severe coronary stenosis. In this study, we found that patients in the experimental group exhibited significantly higher NHR than those in the control group, with a statistically significant difference.

Studies have predominantly focused on the independent evaluation of the severity of coronary artery disease (CAD) through the use of single markers such as NLR or LDL-C/HDL-C, aligning with our findings. However, few studies have explored the combination of inflammatory markers with lipid markers. A three-center, multi-group observational analysis was conducted to evaluate the combined inflammatory and lipid markers NHR and NLR&LDL-C/HDL-C. The results demonstrated that NLR&LDL-C/HDL-C exhibited superior sensitivity and specificity in predicting the severity of CAD. The detected difference was statistically significant. The combination of inflammatory and lipid factors in NLR&LDL-C/HDL-C allows for a more accurate assessment of CAD in ACS patients, enabling early diagnosis and timely risk stratification for appropriate clinical interventions. Furthermore, a predictive model was established (Gensini score = 6.803 + 7.029NLR + 13.079LDL-C/HDL-C, $R^2$ = 0.708).

The study acknowledged that due to the relatively small sample size of this three-center study, the research results may be biased. NHR levels were collected at hospital admission, and the study did not investigate the prognostic value of NHR for adverse cardiovascular outcomes. Additionally, patients in the emergency department may have already received therapeutic interventions, leading to potential unknown and unmeasurable confounding factors such as education level, family income, dietary habits, and exercise habits. Further validation through larger sample studies is necessary.

## Conclusion

Our findings reveal that the combined marker NLR&LDL-C/HDL-C exhibits a stronger association with the severity of coronary artery stenosis compared to NHR. Unlike many other biomarkers, NLR&LDL-C/HDL-C is relatively inexpensive and readily accessible. It offers significant predictive value in assessing the severity of coronary artery stenosis and serves as an effective predictive factor for evaluating coronary artery disease severity in ACS patients.

## Supporting information

**S1 Data.**
(XLSX)

## Acknowledgments

We extend our heartfelt thanks to all individuals who assisted in the preparation of this paper.

## Author Contributions

**Conceptualization:** Shuaishuai Yuan, Lingling Li, Tian Pu, Xizhen Fan, Pailing Xie, Peijun Li.

**Data curation:** Shuaishuai Yuan, Lingling Li, Tian Pu, Xizhen Fan, Zheng Wang, Pailing Xie.

**Formal analysis:** Shuaishuai Yuan, Lingling Li, Tian Pu, Xizhen Fan, Zheng Wang, Pailing Xie.

**Investigation:** Shuaishuai Yuan, Lingling Li, Tian Pu, Xizhen Fan, Zheng Wang, Pailing Xie.

**Methodology:** Shuaishuai Yuan, Lingling Li, Tian Pu, Pailing Xie, Peijun Li.

**Resources:** Shuaishuai Yuan, Lingling Li, Tian Pu, Xizhen Fan, Zheng Wang, Pailing Xie.

**Software:** Shuaishuai Yuan, Lingling Li, Tian Pu, Zheng Wang, Pailing Xie.

**Supervision:** Shuaishuai Yuan, Lingling Li, Tian Pu, Xizhen Fan, Peijun Li.

**Validation:** Shuaishuai Yuan, Lingling Li, Tian Pu, Xizhen Fan.

**Visualization:** Shuaishuai Yuan, Lingling Li, Tian Pu.

**Writing – original draft:** Shuaishuai Yuan, Lingling Li, Tian Pu.

**Writing – review & editing:** Peijun Li.

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
