## [Decision Letter · Decision Letter 0]

10 Oct 2023

PONE-D-23-25566The relationship between NLR, LDL-C/HDL-C, NHR and coronary artery diseasePLOS ONE

Dear Dr. li,

Thank you for submitting your manuscript to PLOS ONE. After careful consideration, we feel that it has merit but does not fully meet PLOS ONE’s publication criteria as it currently stands. Therefore, we invite you to submit a revised version of the manuscript that addresses the points raised during the review process. Please submit your revised manuscript by Nov 24 2023 11:59PM. If you will need more time than this to complete your revisions, please reply to this message or contact the journal office at plosone@plos.org. Please include the following items when submitting your revised manuscript:A rebuttal letter that responds to each point raised by the academic editor and reviewer(s). You should upload this letter as a separate file labeled 'Response to Reviewers'.A marked-up copy of your manuscript that highlights changes made to the original version. You should upload this as a separate file labeled 'Revised Manuscript with Track Changes'.An unmarked version of your revised paper without tracked changes. You should upload this as a separate file labeled 'Manuscript'.

We look forward to receiving your revised manuscript.

Kind regards,

Suyan Tian

Academic Editor

PLOS ONE

Journal Requirements:

https://www.science.gov/topicpages/m/macrophage+cholesterol+homeostasis

https://bmccardiovascdisord.biomedcentral.com/articles/10.1186/s12872-020-01771-z

In your revision ensure you cite all your sources (including your own works), and quote or rephrase any duplicated text outside the methods section. Further consideration is dependent on these concerns being addressed.

**Additional Editor Comments:**

Please address the points raised by two reviewers, particularly please consider more advanced methods to analyze the data (for example, multivariate regression models) and modify the manuscript to improve its soundness.

Reviewers' comments:

Reviewer's Responses to Questions

**Comments to the Author**

1. Is the manuscript technically sound, and do the data support the conclusions?

Reviewer #1: Partly

Reviewer #2: Yes

2. Has the statistical analysis been performed appropriately and rigorously? 

Reviewer #1: No

Reviewer #2: No

3. Have the authors made all data underlying the findings in their manuscript fully available?

Reviewer #1: Yes

Reviewer #2: Yes

4. Is the manuscript presented in an intelligible fashion and written in standard English?

Reviewer #1: Yes

Reviewer #2: Yes

5. Review Comments to the Author

Reviewer #1: This study compares intensity of inflammatory status and basic lipids concentration in patients with ACS. The subject of this interesting study is worth examining but I am afraid that some basic mistakes at the beginning of analysis may cause substantial bias and have significant impact on the findings.

Major concern:

I have many doubts regarding this control group. For me it is very strange that persons with typical symptoms of ACS, in practice STEMI, have normal coronary arteries. It can happen to the patients with NSTEMI or unstable angina but not STEMI cases. What was a reason of ACS with ECG and biochemistry abnormalities? May be you should be more precise describing inclusion criteria. If you put word „or” between three criteria then I can understand. This issue is principal for the further analyzes.

Minor concerns:

1. You did presented and then analyzed all data employing statistical parametric tests. I presume that first you checked them for normality. Please, complete information about method to check continuous variables for normal distribution.

2. A strength of the correlation should be stressed in your analysis. In my opinion if „r” is below 0.2 it means that correlation is very weak or even none. Having pretty large group of patients this was not due to small number of study participants. According to some medical statisticians, „r” exceeding 0.4 should be considered as important/significant. P in this statistics has some value but it is not decisive. The p value measures how likely you would be observed a correlation of this strength (r value) under the null hypothesis.

Reviewer #2: Thank you for possibility to read this interesting paper written by Shuaishuai Yuan et al.

The study concerns a very important topic. I have some comments,

1. please explain all abbreviations in the paper

2. Tab 1 - did all data have normality test passed?

3. The name of Gensini score is improperly used in tables.

4. Did the authors mean to present multivariable or multivariate analysis?

6. PLOS authors have the option to publish the peer review history of their article (what does this mean?). If published, this will include your full peer review and any attached files.

Reviewer #1: No

Reviewer #2: No

---

## [Author Response · Author response to Decision Letter 0]

4 Mar 2024

About Reviewer #1: 

The selection criteria for the article have been modified by adding "or" between the three criteria.

1.A detailed description of the statistical analysis of the article has been provided to ensure its appropriateness and rigor.Kolmogorov–Smirnov test was used to verify that the continuous variable was normally distributed.

2.The strength of correlation has been emphasized in the analysis as required.

About Reviewer #2: 

1.All abbreviations in the text have been explained as required.

2.Table 1- All data were tested for normality using the Kolmogorov Smirnov test method.

3.The name of the Gensini score has been modified in the table.

4.Propose multivariate analysis based on previously published articles to enable the application of more data and improve the accuracy of the results.

---

## [Decision Letter · Decision Letter 1]

22 Mar 2024

PONE-D-23-25566R1NLR、LDL-C/HDL-C、NHR与冠状动脉疾病的关系PLOS ONE

Dear Dr. li,

Thank you for submitting your manuscript to PLOS ONE. After careful consideration, we feel that it has merit but does not fully meet PLOS ONE’s publication criteria as it currently stands. Therefore, we invite you to submit a revised version of the manuscript that addresses the points raised during the review process.

We look forward to receiving your revised manuscript.

Kind regards,

Suyan Tian

Academic Editor

PLOS ONE

Journal Requirements:

**Additional Editor Comments:**

Two minor points caught my attention:

1) Please modify the title, part of which in the submission system are in Chinese. Further, even without this problem, the title fails to hightlight the work the authors have done.

2) Please have the main text English edited by a native speaker.

Reviewers' comments:

Reviewer's Responses to Questions

**Comments to the Author**

1. If the authors have adequately addressed your comments raised in a previous round of review and you feel that this manuscript is now acceptable for publication, you may indicate that here to bypass the “Comments to the Author” section, enter your conflict of interest statement in the “Confidential to Editor” section, and submit your "Accept" recommendation.

Reviewer #1: All comments have been addressed

2. Is the manuscript technically sound, and do the data support the conclusions?

Reviewer #1: Yes

3. Has the statistical analysis been performed appropriately and rigorously? 

Reviewer #1: Yes

4. Have the authors made all data underlying the findings in their manuscript fully available?

Reviewer #1: Yes

5. Is the manuscript presented in an intelligible fashion and written in standard English?

Reviewer #1: Yes

6. Review Comments to the Author

Reviewer #1: Dear Authors,

You have addressed correctly all concerns pointed in my first review. Now your manuscript sounds better than previously.

7. PLOS authors have the option to publish the peer review history of their article (what does this mean?). If published, this will include your full peer review and any attached files.

Reviewer #1: No

---

## [Author Response · Author response to Decision Letter 1]

22 Apr 2024

I have fully addressed the comments raised in the previous round of review and believe that the manuscript can now be published.

---

## [Editor Report · Decision Letter 2]

24 Apr 2024

PONE-D-23-25566R2The relationship between NLR, LDL-C/HDL-C, NHR and coronary artery diseasePLOS ONE

Dear Dr. li,

Thank you for submitting your manuscript to PLOS ONE. After careful consideration, we feel that it has merit but does not fully meet PLOS ONE’s publication criteria as it currently stands. Therefore, we invite you to submit a revised version of the manuscript that addresses the points raised during the review process. Please have the manuscript edited by a native speaker to improve its English.

We look forward to receiving your revised manuscript.

Kind regards,

Suyan Tian

Academic Editor

PLOS ONE
---

## [Author Response · Author response to Decision Letter 2]

29 Apr 2024

We have asked the native English speaker to help us check the grammarly and the clarity of the manuscript.

---

## [Editor Report · Decision Letter 3]

1 May 2024

PONE-D-23-25566R3The relationship between NLR, LDL-C/HDL-C, NHR and coronary artery diseasePLOS ONE

Dear Dr. li,

Thank you for submitting your manuscript to PLOS ONE. After careful consideration, we feel that it has merit but does not fully meet PLOS ONE’s publication criteria as it currently stands. Therefore, we invite you to submit a revised version of the manuscript that addresses the points raised during the review process. Please consider to have a native speaker edited the manuscript to improve its English, otherwise, I have to reject the manuscript.

We look forward to receiving your revised manuscript.

Kind regards,

Suyan Tian

Academic Editor

PLOS ONE
---

## [Author Response · Author response to Decision Letter 3]

23 May 2024

We have asked the native English speaker to help us check the grammarly and the clarity of the manuscript.

---

## [Editor Report · Decision Letter 4]

24 May 2024

The relationship between NLR, LDL-C/HDL-C, NHR and coronary artery disease

PONE-D-23-25566R4

Dear Dr. li,

We’re pleased to inform you that your manuscript has been judged scientifically suitable for publication and will be formally accepted for publication once it meets all outstanding technical requirements.

Kind regards,

Suyan Tian

Academic Editor

PLOS ONE
---

## [Editor Report · Acceptance letter]

29 May 2024

PONE-D-23-25566R4 

PLOS ONE

Dear Dr. Li, 

I'm pleased to inform you that your manuscript has been deemed suitable for publication in PLOS ONE. Congratulations! Your manuscript is now being handed over to our production team.

Kind regards, 

on behalf of

Dr. Suyan Tian 

Academic Editor

PLOS ONE